# FAHFAs Regulate the Proliferation of C2C12 Myoblasts and Induce a Shift toward a More Oxidative Phenotype in Mouse Skeletal Muscle

**DOI:** 10.3390/ijms21239046

**Published:** 2020-11-28

**Authors:** Melha Benlebna, Laurence Balas, Laurence Pessemesse, Béatrice Bonafos, Gilles Fouret, Laura Pavlin, Bénédicte Goustard, Sylvie Gaillet, Thierry Durand, Charles Coudray, Christine Feillet-Coudray, François Casas

**Affiliations:** 1DMEM (Dynamique Musculaire et Métabolisme), INRAE, University Montpellier, 34060 Montpellier, France; benlebna1@gmail.com (M.B.); laurence.pessemesse@inrae.fr (L.P.); beatrice.bonafos@inrae.fr (B.B.); gilles.fouret@inra.fr (G.F.); laura.pavlin@inrae.fr (L.P.); benedicte.goustard@inrae.fr (B.G.); sylvie.gaillet@umontpellier.fr (S.G.); charles.coudray@inrae.fr (C.C.); christine.coudray@inrae.fr (C.F.-C.); 2IBMM (Institut des Biomolecules Max Mousseron), CNRS, ENSCM, University Montpellier, 34093 Montpellier, France; laurence.balas@umontpellier.fr (L.B.); thierry.durand@umontpellier.fr (T.D.)

**Keywords:** FAHFAs, skeletal muscle, C2C12, mitochondrial activity, mice

## Abstract

Branched fatty acid esters of hydroxy fatty acids (FAHFAs) are endogenous lipids reported to have antidiabetic and anti-inflammatory effects. Since skeletal muscle is a major target for insulin, the aim of this study is to explore for the first time the influence of several FAHFAs in C2C12 myoblasts and in skeletal muscle phenotype in mice. Here, we show that eleven FAHFAs belonging to different families inhibit C2C12 myoblast proliferation. In addition, all FAHFAs decreased mitochondrial cytochrome c oxidase activity without affecting reactive oxygen species production and the mitochondrial network. During C2C12 myoblasts differentiation, we found that two of the most active lipids, 9-PAHPA and 9-OAHPA, did not significantly affect the fusion index and the expression of myosin heavy chains. However, we found that three months’ intake of 9-PAHPA or 9-OAHPA in mice increased the expression of more oxidative myosin in skeletal muscle without affecting skeletal muscle mass, number, and mean fiber area, mitochondrial activity, and oxidative stress parameters. In conclusion, our study indicated that the eleven FAHFAs tested decreased the proliferation rate of C2C12 myoblasts, probably through the inhibition of mitochondrial activity. In addition, we found that 9-PAHPA or 9-OAHPA supplementation in mice induced a switch toward a more oxidative contractile phenotype of skeletal muscle. These data suggest that the increase in insulin sensitivity previously described for these two FAHFAs is of muscular origin.

## 1. Introduction

The term FAHFA was coined by the Kahn and Sagathelian teams for the fatty acid ester of hydroxy fatty acids that they recently uncovered in white adipose tissues [1]. Regarding the numerous types of naturally occurring FA and HFA, a large diversity of endogenous FAHFA structures exists [2,3]. Among them, palmitic acid hydroxy stearic acid (PAHSA) family members, in particular, have been studied, and several works have reported antidiabetic and anti-inflammatory effects [1,2,4,5], suggesting that they could have a high therapeutic potential to prevent and/or to treat type 2 diabetes. Yore et al. (2014) demonstrated that acute oral administration of 5-PAHSA in insulin-resistant mice improved glucose tolerance and lowered basal glycemia. In addition, chronic 5-PAHSA treatment improves insulin sensitivity in mice [6]. Furthermore, PAHSAs also reduce inflammatory cytokine production from immune cells and ameliorate adipose inflammation in obese mice [5]. Moreover, acute oral administration of 5-PAHSA and 9-PAHSA protects mice against colitis [4]. Recent works have shown that PAHSAs exert their effects, at least in part, through activation of the G-protein coupled receptors, and notably by GPR120 and GPR40 [1,6,7]. However, recent work also indicates that 5-PAHSA treatment for one month increases insulin resistance and promotes lipid accumulation and inflammatory responses in mice liver [8]. Recently, our work indicated that both 9-PAHPA and 9-OAHPA supplementation increased basal metabolism and enhanced insulin sensitivity in healthy mice [9] and partly counteracted insulin resistance in obese mice [10].

The skeletal muscle plays a crucial role in energy homeostasis and is a major target of insulin action. However, nothing is known about the biological effects of FAHFAs in this tissue. In the present study, our aim is to determine the influence of several FAHFAs previously identified in the food [1,11,12] on skeletal muscle phenotype. For this purpose, we tested the influence of eleven FAHFAs on C2C12 murine myoblasts proliferation. Then, we studied the influence of two of them (9-PAHPA and 9-OAHPA) during C2C12 myoblast differentiation and on muscle phenotype in mice. We showed that the FAHFAs tested decreased the proliferation rate of C2C12 myoblasts. In addition, we found that 9-PAHPA or 9-OAHPA supplementation in mice induced a switch toward a more oxidative contractile phenotype of skeletal muscle.

## 2. Results

### 2.1. The FAHFAs Inhibit C2C12 Myoblasts Proliferation

In this study, we tested eleven FAHFAs (5-OAHPA, 7OAHPA, 9-OAHPA, Omega-OAHPA, 7-OAHSA, 10-OAHSA, 12(Z)-10-OAHC18, 5-PAHPA, 7-PAHPA, 9-PAHPA, and 7-PAHSA) belonging to different families that are present in food of animal or vegetable origin [1,11,12,13] (Figure 1). The biological role of these FAHFAs has never been studied before. These FAHFAs were synthesized, as previously described [14].

Mouse myoblast C2C12 is a useful model for studying the proliferation and differentiation of skeletal muscle cells. We analyzed the influence of these FAHFAs (0.1, 1, and 10 µM) on C2C12 myoblast proliferation. The doses used here are lower than the 20 µM previously used in vitro with 5-PAHSA [1,6,8,15]. Surprisingly, we observed that all FAHFAs inhibited myoblast proliferation at the lowest dose (Figure 2). In addition, we found that these inhibitions were dose-dependent (Figure 2).

### 2.2. The FAHFAs Inhibit Cytochrome C Oxidase Activity during C2C12 Myoblast Proliferation

We have previously shown that myoblast proliferation could be regulated by mitochondrial activity [16,17,18]. Therefore, we hypothesized that FAHFAs could modulate myoblast proliferation by regulating mitochondrial activity. To address this issue, we tested the influence of our FAHFAs during myoblast proliferation on cytochrome c oxidase activity (Complex IV), a key enzymatic activity of the mitochondrial respiratory chain. We found that all FAHFAs tested induced a strong decrease of Complex IV activity (Figure 3A). However, FAHFA treatment of C2C12 myoblasts had no effect on citrate synthase activity, commonly used as a quantitative marker of intact mitochondria [19,20,21] (Figure 3B). Since mitochondria are the most important source of ROS within most mammalian cells, we quantified ROS production using an H2DCF-DA probe. Overall, we showed that FAHFAs did not induce any change in ROS production (Figure 3C).

### 2.3. Influence of FAHFAs on Mitochondrial Network Structure

The impact of FAHFAs on COX activity prompted us to study their influence on the structure of the mitochondrial network. However, we found no change in the mitochondrial network, whatever the FAHFAs studied (Figure 4).

### 2.4. Influence of 9-OAHPA and 9-PAHPA on C2C12 Myoblast Differentiation

For further investigation, we selected two FAHFAs out of the eleven synthetic FAHFAs. Our choice for 9-OAHPA and 9-PAHPA was dictated by (1) the results obtained in C2C12 myoblast proliferation and on mitochondrial activity, and (2) the fact that PAHPA and OAHPA were highly increased in the serum of AG4OX mice (AG4OX mice have increased circulating fatty acids and increased adiposity, yet have lower fasting glycemia and better glucose tolerance compared to controls [22,23,24]) compared to wild-type (WT) animals [1].

After a proliferation phase, the C2C12 myoblasts reach confluence and their terminal differentiation is induced by lowering the serum level in the culture medium. C2C12 cells were grown for 5 days in a differentiation medium, with or without 9-OAHPA or 9-PAHPA (0.1, 1, and 10 µM). We found that 9-OAHPA or 9-PAHPA had no significant effects on myoblast fusion index except for treatment with 0.1 of 9-PAHPAs, where a slight decrease in differentiation was shown (Figure 5A,B). Consistent with this observation, 9-OAHPA or 9-PAHPA had no effect on the expression of myosin heavy chain isoforms (MyHC; Figure 5C) and on the expression of myogenin, a key myogenic factor for differentiation (Figure 5D).

### 2.5. Influence of Long-Term Intake of 9-OAHPA and 9-PAHPA on Skeletal Muscle Phenotype of C57BL/6J Mice

We have previously shown that long-term intake of 9-PAHPA or 9-OAHPA (12 weeks), incorporated into the feed, had no effect on body weight, fat, lean mass, and food intake of C57BL/6J mice [9]. To test whether 9-PAHPA or 9-OAHPA could regulate the physical capacity of mice, we compared exercise performance using weight-matched animals (Figure 6A). We performed an incremental exercise test until exhaustion to determine peak velocity at maximal oxygen consumption (VO_2_max). We found that 9-PAHPA or 9-OAHPA supplementation in mice had no impact on peak velocity (Figure 6A). After euthanasia, we also did not observe any difference in skeletal muscle weights (quadriceps, gastrocnemius, and tibialis; Figure 6B).

Next, we examined the cross-sectional area of the tibialis muscle from our mice. To visualize the muscle fibers, we performed immunostaining for laminin, as previously described [16]. The number and area of tibialis fibers were quantified with Definiens Developer 7.1 software after a NanoZoomer scan (Figure 6C). Quantification of the number and area of tibialis fibers indicated that 9-PAHPA or 9-OAHPA supplementation in mice had no impact on the mean cross-sectional area or total fibers number (Figure 6E,F). Furthermore, analysis of the number of fibers in each range of area also showed no difference with or without FAHFAs (Figure 6D).

To analyze whether the 9-PAHPA or 9-OAHPA supplementation could be associated with a fiber-type switch, we measured the expression of the four adult MyHC transcripts by quantitative PCR. While the expression of MyHCI and MyHCIIb remained unchanged, we found that 9-OAHPA significantly increased the expression of MyHCIIa, whereas 9-PAHPA increased the expression of both MyHCIIa and MyHCIIx (Figure 7A). Then, we measured the mitochondrial respiratory chain activities that determine the metabolic properties of skeletal muscle. We found 9-PAHPA or 9-OAHPA supplementation had no effect on mitochondrial respiratory chain activities, citrate synthase, and β-hydroxyacyl-CoA dehydrogenase (β-HAD, a key mitochondrial enzyme in fatty acid oxidation; Figure 7B,C). In addition, analysis of antioxidant enzymes (SOD, Mn-SOD, catalase, and glutathione peroxidase) and markers of oxidative stress (TBARS and SH-group) revealed no difference with or without FAHFAs (Figure 7D,E).

### 2.6. Expression of G Protein-Coupled Receptors in C2C12 Myoblasts and in Skeletal Muscle

The observation that the studied FAHFAs regulate some important features of C2C12 myoblast in proliferation but not during C2C12 differentiation suggests that 9-PAHPA or 9-OAHPA could act through receptors more strongly expressed during proliferation. Previous studies have demonstrated that some of the biological functions of PAHSAs are mediated through G-protein-coupled receptors (GPCRs) [1,4,6,7]. This data led us to study the mRNA expression of several GPCRs previously identified to bind and be activated by free fatty acids and/or lipid molecules such as PAHSAs. In this aim, we measured the mRNA expression of several GPCRs in C2C12 myoblasts in proliferation or in differentiation state. We found that the expression of GPR40, GPR120, GPR109a, GPR119, and GPR120 remained unchanged during myoblast proliferation or differentiation (Figure 8).

## 3. Discussion

In the present study, the effect of several FAHFAs on C2C12 myoblast proliferation and differentiation was investigated for the first time. We also analyzed the influence of a long-term intake of two major FAHFAs, 9-PAHPA and 9-OAHPA, on the skeletal muscle phenotype of C57Bl6/J mice.

Our data showed that the eleven FAHFAs that belong to different families (OAHPA, OAHSA, 12(Z)-OAHC18, PAHPA, and PAHSA), including several regioisomers in each family, strongly inhibited myoblast proliferation. In addition, they markedly decreased mitochondrial respiratory chain activity as attested by the fall of cytochrome c oxidase activity, ranging between 20% for the lower dose and 63% for the higher dose. However, the number of mitochondria and their morphology and ROS production were not modified by FAHFAs treatment in the proliferating C2C12 cell line. As we have previously demonstrated that myoblast proliferation could be regulated by mitochondrial activity [16,17,18], this set of data suggests that FAHFAs inhibit C2C12 myoblast proliferation through the inhibition of mitochondrial activity. Modulating the myogenic program of the stem cell population within skeletal muscles may be beneficial after an injury or during aging. In fact, after skeletal muscle injury, a regeneration process takes place to repair muscle, and, during aging, muscle repair efficiency is reduced, contributing to sarcopenia [25]. Our results suggest that FAHFAs could regulate skeletal muscle regeneration duration after an injury or during aging. Nevertheless, this hypothesis deserves further investigation.

For the study conducted on C2C12 differentiation and for the study conducted in mice, two major FAHFAs, 9-OAHPA and 9-PAHPA, were used. These were selected based on their abovementioned efficient inhibitory effects on C2C12 myoblast proliferation and mitochondrial activity. Our choice was strengthened by the fact that the level/concentration of these two promising candidates was highly increased [1] in the serum of AG4OX mice, which have lower fasting glycemia and better glucose tolerance compared to controls. After tight control of the proliferation phase, skeletal muscle differentiation is a highly-regulated process that includes several successive steps, such as withdrawal from the cell cycle and the expression of key myogenic factors such as myogenin and fusion into multinucleated myotubes expressing myosin heavy chains. While FAHFAs had significant effects in proliferating C2C12 cells, 9-PAHPA and 9-OAHPA did not significantly affect C2C12 cell differentiation, although they appeared to induce a slight decrease in the fusion index and myogenin expression.

Skeletal muscle contains myofibers differing in contractile function, mitochondrial content, and, consequently, metabolic properties. Slow-twitch fibers are characterized by type I myosin heavy chain (MyHC) expression and a high mitochondrial density, leading to a prominent oxidative metabolism. Fast-twitch fibers express type II MyHCs, including 3 subtypes: IIa, IIx, and IIb. IIb fibers display a reduced mitochondrial density and a major glycolytic metabolism. The oxidative capacity of type IIa and type IIx fibers is intermediate between that recorded for type I and type IIb fibers [26,27]. Interestingly, in the tibialis muscle, we show that 9-OAHPA significantly increased the expression of MyHCIIa, whereas 9-PAHPA increased the expression of both MyHCIIa and MyHCIIx. Since the tibialis muscle expresses less than 1% type I fibers, this increase in contractile IIa and IIx fibers, which have the most oxidative metabolism within this muscle, shows that these two FAHFAs promote a more oxidative contractile type in skeletal muscle. Recently, we demonstrated that the long-term intake of 9-PAHPA or 9-OAHPA increased basal metabolism and insulin sensitivity in mice [9,10]. Skeletal muscle is a major target tissue for insulin because of its mass and metabolic activity. The observation that 9-PAHPA or 9-OAHPA induces a switch toward a more oxidative contractile phenotype is in line with the possibility that these two FAHFAs increase the insulin sensitivity of skeletal muscle. However, the long-term intake of 9-PAHPA or 9-OAHPA (12 weeks) had no effect on the other main features of skeletal muscle phenotype since we observed no change in (1) skeletal muscle mass, (2) size and number of muscle fibers, (3) mitochondrial activity, and (4) oxidative stress parameters. In addition, our data indicate that 9-PAHPA and 9-OAHPA in mice have no deleterious effect on skeletal muscle in healthy mice in contrast to our previous observation on the liver [9].

Bioactive lipids are known to exert their effects through binding to specific receptors, including both the nuclear receptor class and GPCR class. In this study, we investigated the expression of different GPCRs, previously identified to bind and be activated by free fatty acids and/or FAHFAs such as PAHSAs [1,4,6,7], to determine if any of them were more specifically expressed in proliferative myoblasts. However, none of the expression of different GPCRs seemed to change between proliferation and differentiation of myoblasts. These sets of data suggest that the GPCRs analyzed herein did not mediate the biological effects of the eleven FAHFAs tested on proliferating myoblasts. Further investigation will be required to determine the receptor(s) responsible for the effects of these eleven FAHFAs during myoblast proliferation.

For the first time, we have compared FAHFAs belonging to different families in the same study. Our data recorded on C2C12 proliferation indicate that for the OAHPA (5-, 7-, 9-, and omega), PAHPA (5-, 7-, and 9-) and OAHSA (7- and 10-) families, the position of the FAHFA ester function in the regioisomers does not seem to have any effect. In addition, we did not observe any significant difference according to the length of the carbon chain (C16 vs. C18) for either branched fatty acid, or hydroxy fatty acid (for example, 7-OAHPA vs. 7-OAHSA and 7-OAHPA vs. 7-PAHPA). Finally, the monounsaturation of oleic acid did not seem to induce a differential response on C2C12 cell proliferation either. For the in vivo study, we deliberately chose two different fatty acids connected in position 9 with a palmitic acid (9-OAHPA vs. 9-PAHPA). In this study, we found that both 9-OAHPA and 9-PAHPA affected the contractile muscle phenotype in a similar way, whereas our previous results indicated that these two FAHFAs had different actions on the liver. These observations suggest that the length of the chain could induce specific biological effects [9].

## 4. Materials and Methods

### 4.1. FAHFAs Synthesis

Eleven FAHFAs (depicted in Figure 1) were synthetized, as previously described [14]. 5-OAHPA, 7OAHPA, 9-OAHPA, Omega-OAHPA, 7-OAHSA, 10-OAHSA, 12(Z)-10-OAHC18, 5-PAHPA, 7-PAHPA, 9-PAHPA, and 7-PAHSA were dissolved in DMSO.

### 4.2. Cell Culture and Treatments

Mouse myoblasts of the C2C12 cell line (ATCC) were seeded at a plating density of 7000 cells/cm^2^ in 6-well plates. They were grown in DMEM supplemented with gentamycin (50 μg/mL), amphoterecin (50 μg/mL), and fetal calf serum (10%). Terminal differentiation was induced at cell confluence by lowering the medium serum concentration (0.5%).

When indicated, 5-OAHPA, 7OAHPA, 9-OAHPA, Omega-OAHPA, 7-OAHSA, 10-OAHSA, 12(Z)-10-OAHC18, 5-PAHPA, 7-PAHPA, 9-PAHPA, or 7-PAHSA were added to the culture medium at the indicated concentration. For the proliferation studies, FAHFAs were added 24 h after seeding the cells. For the differentiation studies, FAHFAs were added at the confluence at the same time as the change in media condition. Three doses of FAHFAs were studied at a final concentration of 0.1, 1, or 10 µM.

Cell proliferation was measured by total DNA quantification using a QuantiFluor dsDNA system (Promega, WI, USA) according to the manufacturer’s protocol, as described [16].

### 4.3. Measurement of Intracellular Reactive Oxygen Species

Reactive oxygen species (ROS) accumulation was measured using the 2′,7′-dihydrodichlorofluorescein-diacetate (H2DCF-DA) probe (Invitrogen, Thermo Fisher Scientific, Waltham, MA, USA). C2C12 cells grown on 24-well plates were washed with Locke buffer (140 mM NaCl; 5 mM KCl; 1.2 mM MgCl_2_; 1.8 mM CaCl_2_; 10 mM glucose; 10 mM Hepes, 1 M Tris-HCl pH 7.5) and then incubated with 10 µM H2DCF-DA probe in Locke buffer for 20 min at 37 °C. After a quick wash, fluorescence measurement (λex: 485/λem: 530 nm) was performed using a Synergy2 microplate reader (Biotek France, Colmar, France) for 1 h. H2DCF-DA fluorescence was normalized using DNA content, as previously described [28,29].

### 4.4. Cytoimmunofluorescence

Cytoimmunofluorescence myoblast differentiation was assessed by observation of morphological changes and accumulation of muscle-specific markers. After methanol fixation and three washes with PBS-gelatine (0.2%), cells were labeled with an antibody raised against Troponin T (T6277, Sigma, diluted at 1:50)(Sigma-Aldrich, Saint-Quentin Fallavier, France). Nuclei were stained with Hoechst 33258 (1 μg/mL). To assess the extent of differentiation, the fusion index (percentage of nuclei incorporated into myotubes relative to the total number of nuclei) was calculated 120 h after adding the differentiation medium. Nuclei were counted on 10 images/dish using ImageJ software.

To test the potential influence of FAHFAs on the mitochondrial network structure, mitochondria were labeled using 100 nM of Mitotracker^TM^ Red (Molecular Probes, Thermo Fisher Scientific, Waltham, MA, USA).

### 4.5. Protein Levels

Protein levels were assessed by Western blotting. Total proteins were lysed in Tris-NP40 (100 mM Tris, 0.7% NP40, pH 7.4) and measured using the Bio-Rad protein assay (Hercules, CA, USA). Then, 20 μg of protein were electrophoresed onto 10% SDS-PAGE gels and blotted onto a nitrocellulose membrane. Membranes were probed with antibodies raised against myogenin (sc-576, Santa Cruz Biotechnology, diluted at 1:400), all MyHC isoforms (M7523, Sigma, diluted at 1:1000), and α-Tubulin (DM1A, Cell Signaling, diluted at 1:6000). Signals were revealed using a ClarityTM Western ECL Substrate Kit (Hercules, CA, USA), and proteins were visualized by enhanced chemiluminescence using the ChemiDoc Touch Imaging System (Hercules, CA, USA) and quantified with Image Lab™ Touch Software (version 5.2.1) (Hercules, CA, USA).

### 4.6. Gene Expression Studies

Total RNAs were extracted from quadriceps muscle using Trizol^®^, and cDNAs were generated using the PrimeScript™ 1st strand cDNA Synthesis Kit (Takara Bio, Saint-Germain-en-Laye, France). Real-time PCR was performed using SYBR^®^ Premix Ex Taq™ II (Takara Bio) and Applied Biosystems Step-One Plus (Thermo Fisher Scientific, Waltham, MA, USA). Gene expression was normalized to the expression of the housekeeping gene *Rps9* and is expressed as means ± SEM. Student’s *t*-tests were used to determine all *p*-values.

The primers for the determination of MyHC were previously described [30]. For GPCRs expression, the following primers were used: *GPR40* (forward, AGGCGCTCTCCTCACACTC; reverse, CTAGCCACATTGGAGGCATTA); *GPR43* (forward, TGCTCTGAAGAAGCCAATCA; reverse, TTCTCCTCTGGTCCAGTGCT); *GPR109a* (forward, CTGCTCAGGCAGGATCATCT; reverse, CCCTCTTGATCTTGGCATGT); *GPR119* (forward, ATTCCAGCAGACCACCTACCA; reverse, GCACAAACCTTGGGTGAAACA); *GPR120* (forward, TTGGTGTTGAGCGTCGTG; reverse, CCAGCAGTGAGACGACAAAG).

The PCR overall efficiency (E) was calculated from the slopes of the standard curves according to the equation E = [10^(−1/slope)^] − 1, and this value was higher than 95% for all assays. The relative abundance of each sample was then normalized according to the equation: relative quantity = 2^−ΔΔCt^. All of the experiments were performed according to the minimum information for publication of quantitative real-time PCR experiment (MIQE) guidelines [31,32].

### 4.7. Mitochondrial Enzymatic Activities

Mitochondrial respiratory complex activities were determined as previously described [30]. Citrate synthase (CS) activity was measured according to Srere [33] by following the color formation of 5-thio-2-nitrobenzoic acid at 412 nm. The enzymatic activity of complex IV (cytochrome c oxidase) was measured according to Wharton and Tzagoloff [34]. The activity of cytochrome oxidase is measured with a spectrophotometer following the oxidation of reduced cytochrome c at 550 nm, pH 7.4, and 37 °C. Cytochrome c was reduced with sodium dithionite.

### 4.8. Animals and Ethics Statement

Thirty 6-week-old male C57BL/6J mice (Charles River, L’Arbresle, France), weighing about 22 g, were housed (5 per cage) under conditions of constant temperature (20–22 °C), humidity (45–50%), and a standard dark cycle (20.00–08.00 h). The mice were randomized into three groups of ten animals, as previously described, according to their initial weight [9]. They were fed for 12 weeks with one of the three following semipurified diets: (1) control diet, (2) control diet + 9-PAHPA, and (3) control diet + 9-OAHPA. This corresponds to a FAHFA intake of about 30 µmol/day/kg of mice (i.e., about 15 mg/day/kg). Male mice were housed and maintained on a 12-h light/dark cycle (lights on at 7:00 pm). Our institution guidelines for the care and use of laboratory animals were observed, and all experimental procedures were approved by the local ethical committee in Montpellier, France (Reference APAFIS#12759-2017121912214385). See Benlebna and co-workers [9].

### 4.9. Treadmill Studies

Treadmill exercise tests were performed two weeks before the end of the study using a motorized rodent treadmill (Exer-6M Open Treadmill; Columbus Instruments, OH, USA). Before the exercise performance test, the mice were accustomed to the treadmill with a 5-min run at 7 m/min for 2 days. To evaluate the maximal aerobic speed, as previously described [35], the exercise test regimen was 10 m/min for the first minute, followed by 2 m/min increments every 2 min until 14 m/min, then 1 m/min every minute.

### 4.10. Histological Studies

After euthanasia by cervical dislocation, the tibialis anterior muscles were collected, freshly frozen in Tissue-Tek (Microm Microtech, Brignais, France), and then stored at −80 °C. For immunostaining, the sections were fixed in PBS, 4% PFA at room temperature (RT) for 5 min, permeabilized 30 min in PBS, 20% horse serum, and 0.1% triton at RT, and incubated with the anti-laminin (1/200, rabbit polyclonal, Sigma L9393) in a solution of PBS, 20% goat serum, and 1% BSA for 24 h at 4 °C. Sections were washed in PBS 3X for 10 min and incubated with the secondary antibody in PBS for 1 h at 37 °C. Sections were washed in PBS 2X for 10 min, incubated 30 s with DAPI, and washed once in PBS for 10 min and mounted.

For the morphometric analysis, muscle sections were scanned using a NanoZoomer (Hamamatsu Photonics, Massy, France) with a 20X objective. Definiens developer 7.1. software was used to analyze and quantify the pictures for each entire area.

### 4.11. Oxidative Stress

Quadriceps were homogenized in phosphate buffer (50 mM, pH 7), 1 g for 9 mL buffer, using a Polytron homogenizer. The thiobarbituric acid reactive substances (TBARS) were measured in homogenate according to the methods of Sunderman [36]. The remaining homogenate was centrifuged at 1000× *g* for 10 min at 4 °C, and the supernatant was used for the other analyses of oxidative stress and mitochondrial parameters, as previously described [9].

### 4.12. Statistical Analyses

Statistical analysis of the in vitro study: the results are expressed as means ± SEM; the effect of the groups, including the control group, was tested by one-way ANOVA, followed up by a Fisher’s least significant difference (LSD) test; all the groups, except the control group, were also tested for the effects of dose, FAHFAs, and their interaction by two-way ANOVA test. Statistical analysis of the in vivo study: the results are expressed as means ± SEM or as percentages. Statistical significances of the differences between groups were evaluated with Student’s *t*-tests.

## 5. Conclusions

In conclusion, we found, for the first time, that 11 FAHFAs belonging to different families that are present in food of animal or vegetable origin decreased the proliferation rate of C2C12 myoblasts, probably through the inhibition of mitochondrial activity. In addition, we did not observe any specificity of action of FAHFAs since all those tested inhibited cell proliferation. Moreover, despite the fact that 9-PAHPA and 9-OAHPA had little or no biological effects during myoblast differentiation, we found that 12 weeks of 9-PAHPA or 9-OAHPA supplementation in mice induced a switch toward a more oxidative contractile phenotype of skeletal muscle. These data suggest that the increase in insulin sensitivity previously described for these 2 FAHFAs is of muscular origin [9]. Overall, these results highlighted the complexity of action of FAHFAs, whose biological functions seem to depend on their structure and target tissues.

## Figures and Tables

**Figure 1 ijms-21-09046-f001:**
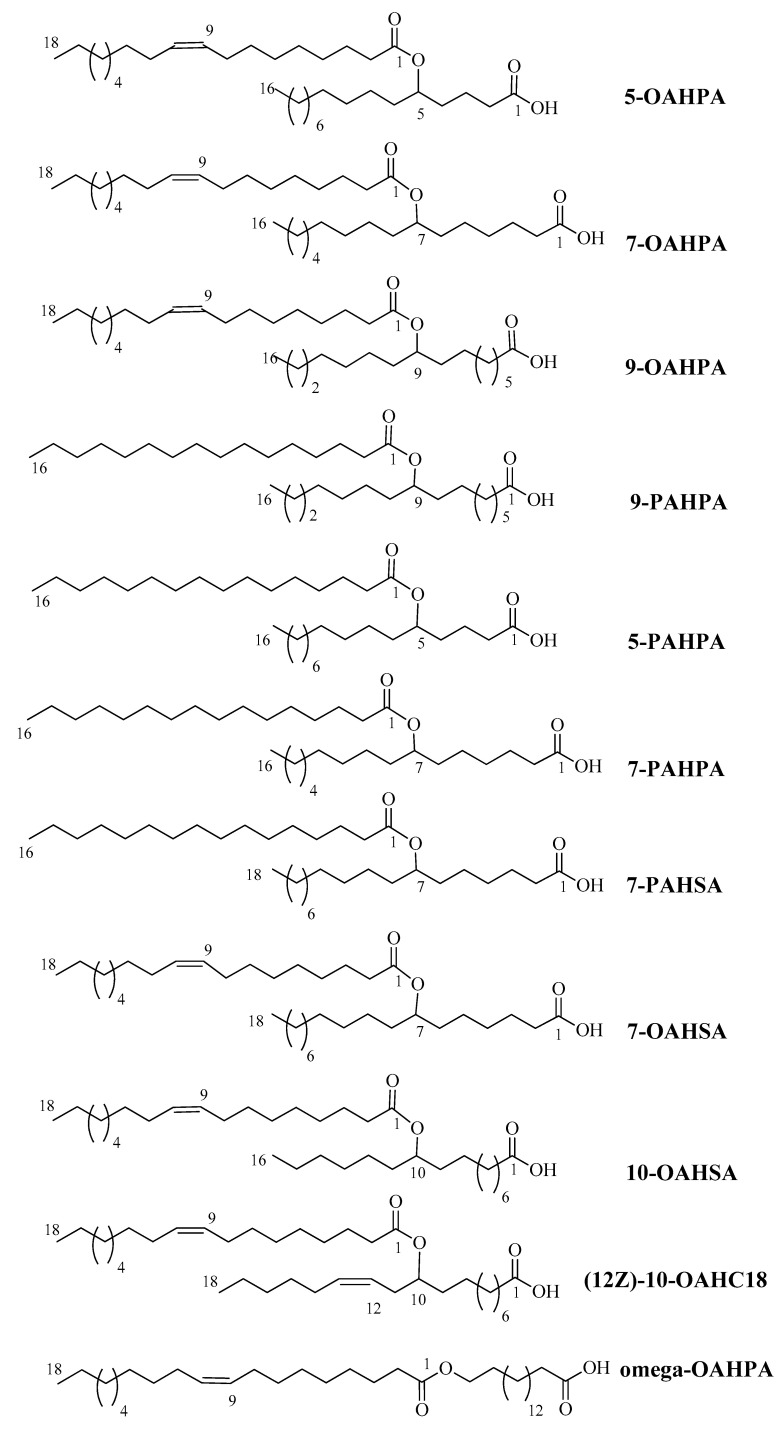
Molecular formula, mass, and names of the fatty acid esters of hydroxy fatty acids (FAHFAs) used in this study.

**Figure 2 ijms-21-09046-f002:**
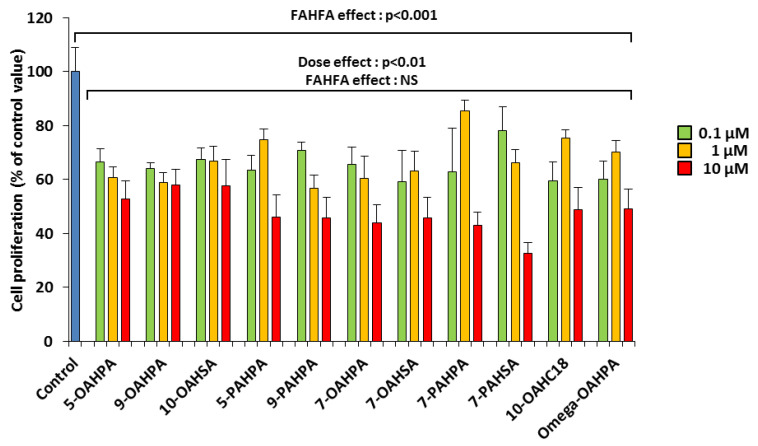
FAHFAs decreased C2C12 myoblast proliferation. Cell proliferation was measured by total DNA quantification using QuantiFluor 24 h after FAHFA treatment at the indicated concentration (*n* = 6 for each group). The effect on the groups, including the control group, was tested by one-way ANOVA, followed up by a Fisher’s least significant difference (LSD) test; all the groups, except the control group, were also tested for the effects of dose, FAHFAs, and their interaction by two-way ANOVA test. Results are expressed as means ± SEM.

**Figure 3 ijms-21-09046-f003:**
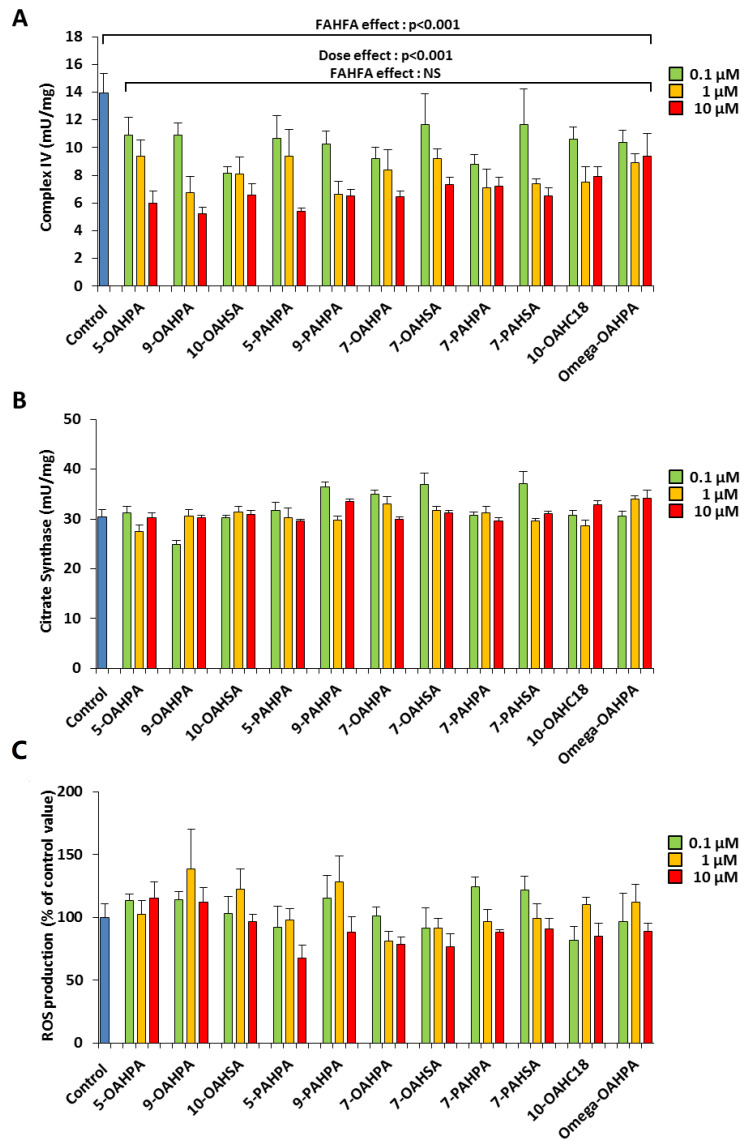
FAHFAs decreased mitochondrial respiratory chain activity in proliferating C2C12 myoblasts. (**A**) FAHFAs decreased mitochondrial complex IV activity in proliferating C2C12 myoblasts. Enzymatic activity was measured 24 h after FAHFA treatment at the indicated concentration (*n* = 6 for each group). (**B**) FAHFAs had no influence on citrate synthase activity in proliferating C2C12 myoblasts. Enzymatic activity was measured 24 h after FAHFA treatment at the indicated concentration (*n* = 6 for each group). (**C**) FAHFAs had no influence on ROS production in proliferating C2C12 myoblasts. ROS accumulation was measured using the H2DCF-DA probe 24 h after FAHFA treatment at the indicated concentration (*n* = 6 for each group). The effect on the groups, including the control group, was tested by one-way ANOVA, followed up by a Fisher’s least significant difference (LSD) test; all the groups, except the control group, were also tested for the effects of dose, FAHFAs, and their interaction by two-way ANOVA test. Results are expressed as means ± SEM.

**Figure 4 ijms-21-09046-f004:**
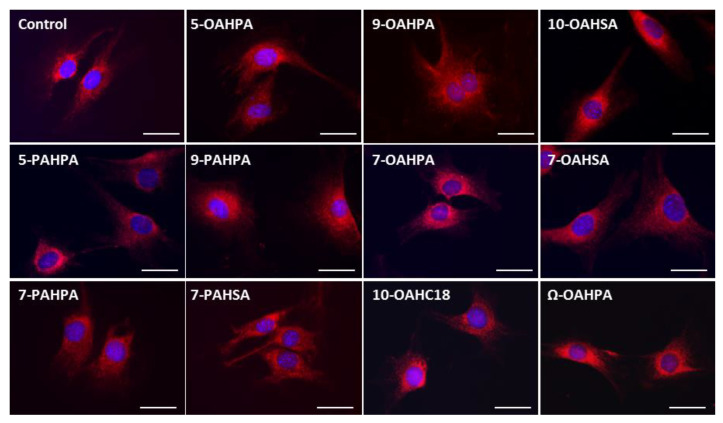
Influence of FAHFAs on mitochondrial network structure. The mitochondrial network structure was visualized using Mitotracker^TM^ Red. Scale Bar, 20 µm.

**Figure 5 ijms-21-09046-f005:**
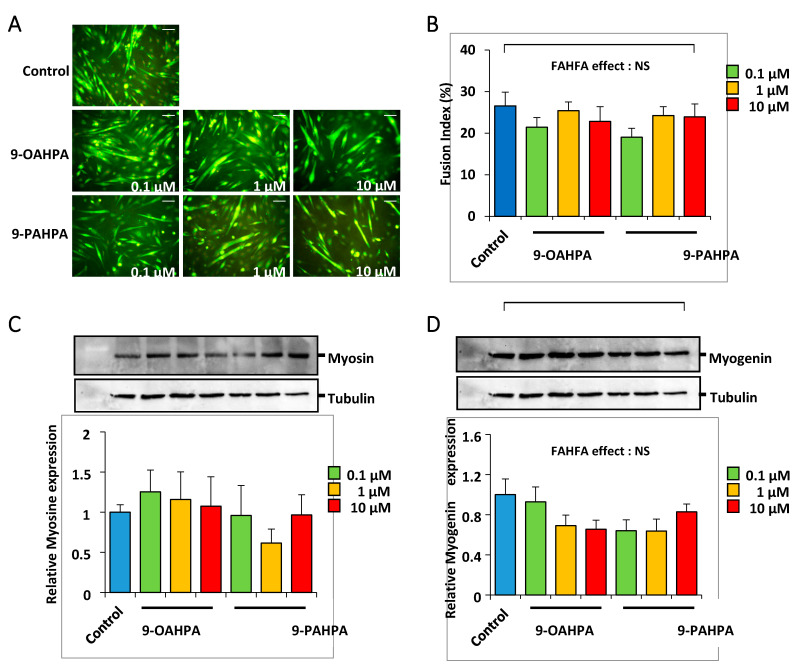
Influence of 9-OAHPA and 9-PAHPA on C2C12 myoblast differentiation. (**A**) Cytoimmunofluorescence studies using antibodies raised against Troponin T were performed on C2C12 myoblasts after 5 days in differentiation medium. FAHFAs were added at the induction of the differentiation at the indicated concentration (*n* = 6 for each group). Scale bar, 200 μm. (**B**) The extent of differentiation was assessed by the fusion index (percentage of nuclei incorporated into myotubes relative to the total number of nuclei). (**C**,**D**) Total MyHC and myogenin protein levels were analyzed by Western blot (*n* = 6 for each group). Typical blots are shown. Proteins were quantified with Image Lab™ Touch Software. There is no statistical difference (one-way ANOVA, followed-up by an LSD test). Results are expressed as means ± SEM.

**Figure 6 ijms-21-09046-f006:**
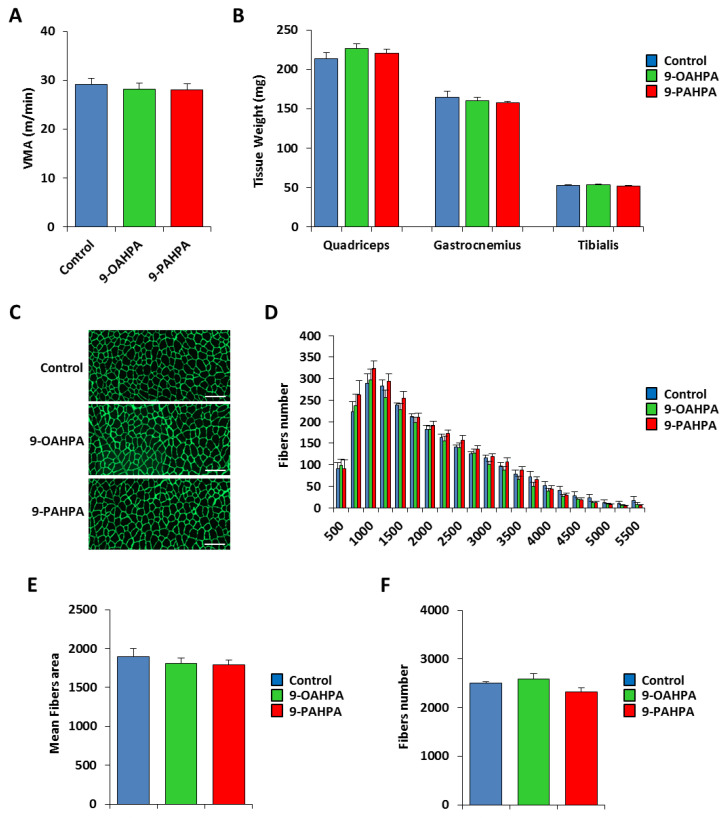
Influence of long-term intake of 9-OAHPA and 9-PAHPA on skeletal muscle phenotype of C57BL/6J mice. (**A**) Maximal aerobic speed (*n* = 10 for each group). (**B**) Quadriceps, gastrocnemius, and tibialis muscles weight (*n* = 10 for each group). (**C**) Representative anti-laminin staining on cryosections of tibialis muscles. Scale bar, 200 μm. (**D**) Fiber size distribution in tibialis muscles (*n* = 10 for each group). (**E**) Mean fibers area (*n* = 10 for each group). (**F**) Total fibers number (*n* = 10 for each group). There is no statistical difference (Student’s *t*-test). Results are expressed as means ± SEM.

**Figure 7 ijms-21-09046-f007:**
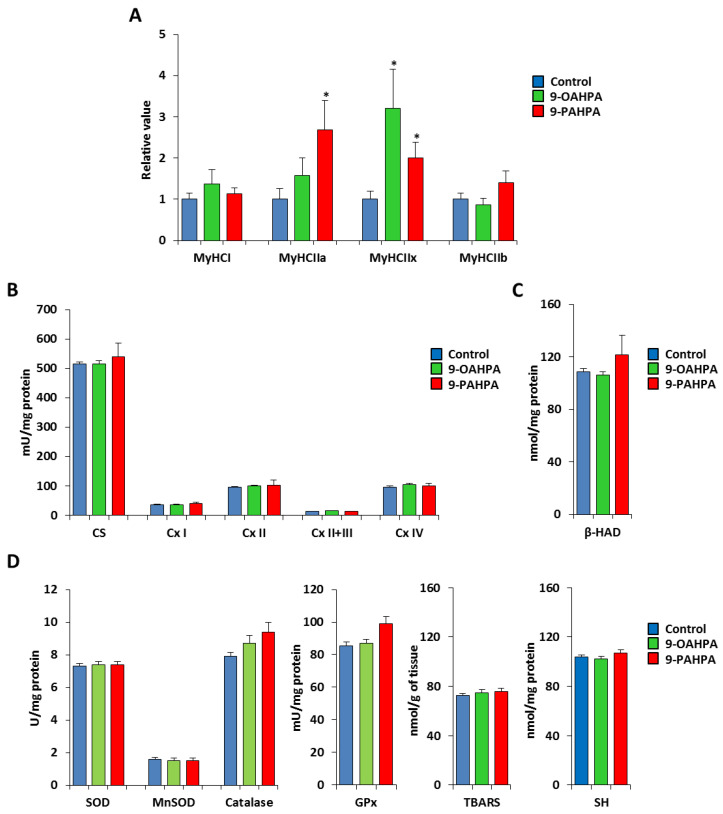
Influence of long-term intake of 9-OAHPA and 9-PAHPA on metabolic and contractile features of skeletal muscle. (**A**) Relative mRNA expression levels of the 4 adult MyHC isoforms in the tibialis muscle (*n* = 10 for each group). (**B**) Mitochondrial enzymatic activities (n = 10 for each group). (**C**) β-HAD enzymatic activity (*n* = 10 for each group). (**D**) Analysis of antioxidant enzymes (SOD, Mn-SOD, catalase, and glutathione peroxidase) and markers of oxidative stress (TBARS and SH-group; *n* = 10 for each group). SOD, total superoxide dismutase; MnSOD, manganese superoxide dismutase; GPx, glutathione peroxidase; SH, thiols; TBARS, thiobarbituric acid. Statistical significance: * *p* < 0.05 vs. control. Student’s *t*-test. Results are expressed as means ± SEM.

**Figure 8 ijms-21-09046-f008:**
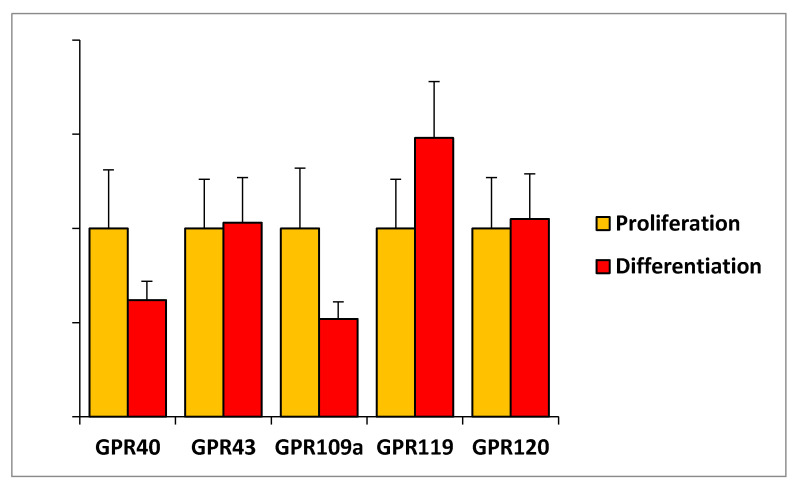
Expression of G-protein-coupled receptors in C2C12 myoblasts. Relative mRNA expression levels of several indicated GPCRs in C2C12 myoblasts in proliferation or in differentiation (*n* = 8 for each group). Relative mRNA levels obtained are expressed in percent of the corresponding control value in proliferation. Results are expressed as means ± SEM.

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
