# Peer review of "FAHFAs Regulate the Proliferation of C2C12 Myoblasts and Induce a Shift toward a More Oxidative Phenotype in Mouse Skeletal Muscle"

_ijms, 2020, doi:10.3390/ijms21239046_

Round 1

Reviewer 1 Report

The study of the impact of FAHFAs on skeletal muscle phenotype is very important to understand its effect on insulin resistance. The study is very interesting and include different FAHFAs belonging to different families and their effect on C2C12 murine myoblasts differentiation and also on muscle phenotype in mice. The weak part, however, is the measurement of mitochondrial function.

The part on mitochondrial function is weak, and does not really address the integrative function of the mitochondria. And even if the treatment show short term effect on complex IV, the information gain by a more integrative assessment of the mitochondrial function would have been easier to correlate with oxidative stress and contractile properties. Furthermore, as the change in complex IV is the only significant change occurring (other than cell proliferation), it would be good to have more information on the method used. Without a proper blank, the assay is not accurate.

Author Response

Response:

I agree with the reviewer that technically the study of mitochondrial function could have been more thorough at the cellular level. However, we have studied the activity of two of the main enzymes that control the activity of the mitochondrial respiratory chain: 1) citrate synthase, a key enzyme of the krebs cycle; 2) complex IV which catalyzes the transfer of electrons to an oxygen molecule. Indeed, by using an Oroboros we would have had a more complete vision of the mitochondrial function. However, in this study, during the proliferation phase, we compared untreated C2C12 cells with cells treated with 11 different FAHFAs at 3 different doses. This represents 34 conditions with 6 samples per condition for a total of 204 samples. With such a quantity of samples the use of Oroboros is not feasible and the spectrophotometric measurement of citrate synthase and IV complex activity seemed to be the best compromise for us.

During proliferation we did not observe any difference in the production of ROS in cells treated with FAHFAs. During proliferation, the contractile proteins are not yet expressed, therefore it was not possible to perform any correlation between oxidative stress and contractile properties. 

The methods used to determine the activities of citrate synthase (Srere, 1967) and complex IV (Wharton and Tzagaloff) are two internationally recognized methods that have already been cited more than 2000 times. The activity of cytochrome oxidase is measured with a spectrophotometer following the oxidation of reduced cytochrome c at 550nm, pH7.4 and 37°C. Cytochrome c is reduced with sodium dithionite.

  What do you want to do ? New mailCopy  

Reviewer 2 Report

In this manuscript, the authors analyzed the influence of several FAHFAs in C2C12 myoblasts and in skeletal muscle phenotype in mice. They hypothesized that FAHFAs could modulate myoblasts proliferation by regulating mitochondrial activity. They found that elevated FAHFAs inhibit C2C12 myoblast proliferation, decrease mitochondrial cytochrome c oxidase activity without affect reactive oxygen species production and the mitochondrial network. They also found that 9-PAHPA or 9-OAHPA supplementation in mice, induced a switch toward a more oxidative contractile phenotype of skeletal muscle.

The results are interesting; they are a continuation of previous work by the group where they have seen that mitochondrial activity regulates myoblast differentiation.

I have some minor questions:

-FAHFAs are present at low levels in serum and WAT in mice and humans, but it is not clear if some FAHFA family is present in muscle. Have the authors analyzed that FAHFA family is expressed in muscle?

- Indicate how the concentrations to be used of FAHFA in cells and animals were selected?

- Indicate number of repetitions and samples analyzed in the experiments. It can be added in the figure captions or in material and methods.

- After the mice were treated with the two FAHPAs, was their concentration accumulated or changed in the liver or in the white and / or brown fat?

Author Response

Response:

In this study we used 3 different concentrations of FAHFA (0.1, 1 and 10 µM). These concentrations were lower than the 20 µM of 5-PAHSA used on cell lines as previously described (Yore et al (2014); Pflimlin et al (2018); Syed et al (2018) and Wang et al (2018)). This was added in the manuscript. The number of samples analyzed (n=6) is indicated in figures legends.

Unfortunately, we did not analyze FAHFAs in liver and adipose tissue. However, an analysis was performed in mouse muscle. In mice supplemented with 9-PAHPA the concentration of 9-PAHPA increased from 7 to 7000 µg/g and in animals supplemented with 9-OAHPA the concentration of 9-OAHPA increased from 67 to nearly 3000 µg/g. These results, which need to be confirmed on a larger number of samples, nevertheless show that supplementation with FAHFAs is well reflected in the tissues.

  What do you want to do ? New mailCopy  

Reviewer 3 Report

  1. This manuscript thoroughly needs an English edition as it is very poor in most of the paragraphs.
  2. Abstract: starts with “anti-diabetic and anti-inflammation” no logical clue to the findings. It is vague for readers to understand whether the reduction of proliferation and all other consequent findings indicate anti-diabetes effect. The abstract must undergo re-formed and re-phrased. It must end to a conclusion of the findings.
  3. Abstract: please mature your finding with concluding remarks. For instance, this is quite ambiguous for readers “induced a switch toward a more oxidative contractile phenotype of skeletal muscle”.
  4. Lines 17 and 22: without affecting…
  5. Line 43-45: English is poor and it needs re-phrase.
  6. Line 51-57: please re-phrase the paragraph avoids too detailed methodology. This paragraph must indicate the aim/approach/finding.
  7. Lines 68-71: please add a notion of the relation of proliferation assay to the antagonistic effect of these fatty acids.
  8. Regarding statistical analysis (lines 375-376): All statistical significances must be tested for One-way ANOVA Posthoc (e.g. Bonferroni)-corrected multiple comparisons or student paired t-test. However, all analyses were computed with a bare t-test which jeopardizes committing errors. Authors have to re-analyze the data with the right procedure, but otherwise, the data is not trustworthy.
  9. After fulfilling the right statistical analyses, please correct all figure legends and mention them in the last sentence.
  10. What is “as ±sem” do you mean: Means±SEM? Please comply with the consensus agreement for statistical markers.
  11. Line 8: what do you mean “influence” if effect, please change all to “effect” across the MS.
  12. Lines 259-265: This conclusion paragraph is too poor and it does not reflect the importance of your findings. Please re-phrase it and highlight it very clearly.
  13. Figure legends: must be descriptive as it expresses a bit approach and finding followed by statistical analysis method.
  14. Figures: Please apply the same order to all figures: Means±SEM or Means±SD? Axis and caption size and font.
  15. In experiments across the MS: please define the control: I suppose it “untreated”
  16. For instance, Figure 7 A and Figure 8 must be changed from “relative value” to “relative mRNA level”, the legend must be re-phrased.

Author Response

  1. This manuscript thoroughly needs an English edition as it is very poor in most of the paragraphs.

Response: according to the advice of the reviewer, an in-depth proofreading by an english-speaking person has been done

  1. Abstract: starts with “anti-diabetic and anti-inflammation” no logical clue to the findings. It is vague for readers to understand whether the reduction of proliferation and all other consequent findings indicate anti-diabetes effect. The abstract must undergo re-formed and re-phrased. It must end to a conclusion of the findings.

Response: The abstract has been re-formed and re-phrased according to the reviewer advice.

  1. Abstract: please mature your finding with concluding remarks. For instance, this is quite ambiguous for readers “induced a switch toward a more oxidative contractile phenotype of skeletal muscle”.

Response: As a specialist in muscle metabolism this sentence does not seem ambiguous. However, we understand that this may not be clear to non-specialists. An explanation will be brought into the discussion. In tibialis muscle, we show that 9-OAHPA significantly increased the expression of MyHCIIa whereas 9-PAHPA increased the expression of both MyHCIIa and MyHCIIx. Since the tibialis muscle expresses less than 1% type I fibers, this increase in contractile fibers IIa and IIx, which have the most oxidative metabolism within this muscle, shows that these two FAHFAs promote a more oxidative contractile type in the skeletal muscle.

  1. Lines 17 and 22: without affecting…

Response: OK

  1. Line 43-45: English is poor and it needs re-phrase.

Response: The sentence has been rephrased according to the reviewer advice.

  1. Line 51-57: please re-phrase the paragraph avoids too detailed methodology. This paragraph must indicate the aim/approach/finding.

Response: The sentence has been rephrased and the paragraph restructured according to the reviewer advice.

  1. Lines 68-71: please add a notion of the relation of proliferation assay to the antagonistic effect of these fatty acids.

Response: This possible relationship is indicated in the first sentence of the following paragraph: “We have previously shown that myoblasts proliferation could be regulated by mitochondrial activity [16-18]. Therefore, we hypothesized that FAHFAs could modulate myoblasts proliferation by regulating mitochondrial activity”.

  1. Regarding statistical analysis (lines 375-376): All statistical significances must be tested for One-way ANOVA Posthoc (e.g. Bonferroni)-corrected multiple comparisons or student paired t-test. However, all analyses were computed with a bare t-test which jeopardizes committing errors. Authors have to re-analyze the data with the right procedure, but otherwise, the data is not trustworthy.

Response: Data were re-analysed with one-Way or two-Way ANOVA test, according to the procedure written below. However, student paired t-test were not performed as the design of the study does not allow the utilization of this test.

Statistical analysis of in vitro study: results were expressed as means ± SEM; the effect of groups including control group was tested for one-way ANOVA followed up by a Fisher's Least Significant Difference (LSD) test; all the groups except control group were also tested for the effects of dose, FAHFAs, and their interaction by two-way ANOVA test. Statistical analysis of in vivo study: results were expressed as means ± SEM, or as percentages. Statistical significances of the differences between groups were evaluated with Student’s t-test.

  1. After fulfilling the right statistical analyses, please correct all figure legends and mention them in the last sentence.

Response: OK, we did that.

  1. What is “as ±sem” do you mean: Means±SEM? Please comply with the consensus agreement for statistical markers.

Response: The correction has been done.

  1. Line 8: what do you mean “influence” if effect, please change all to “effect” across the MS.

Response: OK

  1. Lines 259-265: This conclusion paragraph is too poor and it does not reflect the importance of your findings. Please re-phrase it and highlight it very clearly.

Response: The conclusion has been reformulated according to the reviewer advice.

  1. Figure legends: must be descriptive as it expresses a bit approach and finding followed by statistical analysis method.

Response: OK, we did that.

  1. Figures: Please apply the same order to all figures: Means±SEM or Means±SD? Axis and caption size and font.

Response: OK, we did that.

  1. In experiments across the MS: please define the control: I suppose it “untreated”

Response: OK

  1. For instance, Figure 7 A and Figure 8 must be changed from “relative value” to “relative mRNA level”, the legend must be re-phrased.

Response: OK , we did that.

  What do you want to do ? New mailCopy  

Round 2

Reviewer 3 Report

Thank you for revising your manuscript as it seems much improved according to the previous comments.